# Farmers’ Perspectives of the Benefits and Risks in Precision Livestock Farming in the EU Pig and Poultry Sectors

**DOI:** 10.3390/ani13182868

**Published:** 2023-09-09

**Authors:** Idan Kopler, Uri Marchaim, Ildikó E. Tikász, Sebastian Opaliński, Eugen Kokin, Kevin Mallinger, Thomas Neubauer, Stefan Gunnarsson, Claus Soerensen, Clive J. C. Phillips, Thomas Banhazi

**Affiliations:** 1European Wing Unit, Galilee Research Institute, Kiryat Shmona 11016, Israel; uri@migal.org.il; 2Agricultural Economics Directorate, Institute of Agricultural Economics, H-1093 Budapest, Hungary; tikasz.ildiko.edit@aki.gov.hu; 3Department of Environmental Hygiene and Animal Welfare, Wroclaw University of Environmental and Life Sciences, 50-375 Wrocław, Poland; sebastian.opalinski@upwr.edu.pl; 4Institute of Forestry and Engineering, Estonian University of Life Science, 51014 Tartu, Estonia; eugen.kokin@emu.ee (E.K.); clive.phillips@curtin.edu.au (C.J.C.P.); 5SBA Research, 1040 Vienna, Austria; kmallinger@sba-research.org (K.M.); tneubauer@sba-research.org (T.N.); 6Department of Animal Environment and Health, Swedish University of Agricultural Sciences, SE-532 23 Skara, Sweden; stefan.gunnarsson@slu.se; 7Department of Electrical and Computer Engineering, Aarhus University, 8000 Aarhus, Denmark; claus.soerensen@ece.au.dk; 8CUSP Institute, Curtin University, Bentley, WA 6102, Australia; 9AgHiTech Kft, H-1101 Budapest, Hungary; thomas.banhazi@plfag.com; 10International College, National Taiwan University, Taipei 10617, Taiwan

**Keywords:** PLF, ICT, livestock, technology-adoption, farmers’ engagement, farmers’ adoption, animal welfare, digitalization

## Abstract

**Simple Summary:**

Smart farming is a concept of agricultural innovation that combines technological, social, economic and institutional changes. It employs novel practices of technologies and farm management at various levels (specifically with a focus on the system perspective) and scales of agricultural production, helping the industry meet the challenges stemming from immense food production demands, environmental impact mitigation and reductions in the workforce. Precision Livestock Farming (PLF) systems will help the industry meet consumer expectations for more environmentally and welfare-friendly production. However, the overwhelming majority of these new technologies originate from outside the farm sector. The adoption of new technologies is affected by the development, dissemination and application of new methodologies, technologies and regulations at the farm level, as well as quantified business models. Subsequently, the utilization of PLF in the pig and especially the poultry sectors should be advocated (the latter due to the foreseen increase in meat production). Therefore, more significant research efforts than those that currently exist are mainly required in the poultry industry. The investigation of farmers’ attitudes and concerns about the acceptance of technological solutions in the livestock sector should be integrally incorporated into any technological development.

**Abstract:**

More efficient livestock production systems are necessary, considering that only 41% of global meat demand will be met by 2050. Moreover, the COVID-19 pandemic crisis has clearly illustrated the necessity of building sustainable and stable agri-food systems. Precision Livestock Farming (PLF) offers the continuous capacity of agriculture to contribute to overall human and animal welfare by providing sufficient goods and services through the application of technical innovations like digitalization. However, adopting new technologies is a challenging issue for farmers, extension services, agri-business and policymakers. We present a review of operational concepts and technological solutions in the pig and poultry sectors, as reflected in 41 and 16 European projects from the last decade, respectively. The European trend of increasing broiler-meat production, which is soon to outpace pork, stresses the need for more outstanding research efforts in the poultry industry. We further present a review of farmers’ attitudes and obstacles to the acceptance of technological solutions in the pig and poultry sectors using examples and lessons learned from recent European projects. Despite the low resonance at the research level, the investigation of farmers’ attitudes and concerns regarding the acceptance of technological solutions in the livestock sector should be incorporated into any technological development.

## 1. Introduction

The challenge of food production in the 21st century may materialize, as the world’s population is expected to increase to 9.4–10.1 billion people by 2050 [1], implying an increase of at least 35%. To meet the demands for increased food production, a significant increase in the number of livestock is expected, especially in the BRIC countries (Brazil, Russia, India, and China) [2]. Moreover, the livestock sector plays a significant economic and social role in the European Union (EU), which accounts for 4.1 million livestock in farms and 36% of the total agricultural activity [3]. However, according to a Deloitte^©^ discussion paper [4], if global warming is to be kept within 2 °C above pre-industrial levels, which requires the emissions associated with the production of meat to be decreased to 3.2 Gt by 2050, only 41% of global meat demand can be met by this date. If global warming is not restricted, heat stress will have increasingly adverse effects on meat and milk production, particularly in developing countries exposed to high temperatures [5,6].

Anthropologically induced environmental changes place constant pressure on animal production due to new and re-emerging pathogens resulting from the natural evolution of microorganisms. This could also potentially reduce the ability of farming communities to develop new crops in already deteriorated ecosystems. Likewise, growing urbanization reduces the labor force availability in areas typically involved in food production, increases costs and reduces the sector’s productive capacity [7]. Moreover, the recent COVID-19 pandemic crisis has clearly illustrated the emerging necessity of building a sustainable and stable agriculture that can sustain its resilience and secure reliable food supplies both regionally and globally amidst a global critical situation.

Agriculture is increasingly becoming knowledge-intensive, digitalized and influenced by technological developments at the supplier and consumer levels [8]. The overwhelming majority of these new technologies originate from outside the farm sector. The adoption of new technologies is affected by the development, dissemination and application of new methodologies, technologies and regulations at the farm level, as well as quantified business models, all of which have implications for farm capital and other inputs. Additionally, farmers’ collective knowledge derives from the knowledge of the individual farmers or stock people, which in turn reflects their training, acquired advice and information. All of these aspects make the adoption of technologies for sustainable farming systems a challenging and dynamic issue for farmers, extension services, agri-business and policymakers. Considering the wide range of objectives related to new technology adaption in the context of livestock farming, it is necessary for farmers, scientists and companies to work together collaboratively.

Smart farming is a concept in agricultural innovation that combines technological, social, economic and institutional changes [9]. It employs novel practices of farm management at various levels (specifically focused on the system level) and scales of agricultural production, helping the industry to meet the challenges stemming from the growing food production demands and reduction in the workforce [10]. The approach of Precision Livestock Farming (PLF) for a sustainable farming system refers to the continuous capacity of agriculture to contribute to overall human and animal welfare by using available information more effectively on farms. In turn, the better utilization of information enables farmers to provide sufficient goods and services in ways that are economically efficient and socially and environmentally responsible [11,12].

PLF uses smart farming technology, which includes the utilization of various types of sensors to collect data, which is thereafter usually transferred collectively by communication networks to servers using Information and Communications Technology (ICT). In this Internet of Things (IoT), by generally accepted definition, large amounts of data from interconnected devices are recorded and analyzed by management information systems, data analysis solutions [13,14] and data analytics [15] domains. The use of the data provided by smart farming potentially helps boost productivity and minimize waste by allowing the necessary actions to be carried out at the right time and in the right place [16]. An FAO report [17] highlighted the importance of ICT as a tool to help meet future food and feed requirements.

Digital technologies have been developed to continuously track real-time production performance and environmental conditions in various livestock facilities [18]. In this sense, they facilitate an improved response to humans’ and animals’ needs by (a) maximizing production efficiency, (b) increasing product quality, (c) improving animal health and welfare, (d) reducing human occupational health and safety risk and (e) mitigating emissions from livestock. Policymakers can also benefit from increased information sharing, which allows them to gather a more complete overview of the situation at the national and regional levels. An additional major benefit connected with ICT use lies in the potential to reach all the layers of society. Moreover, recent technological developments in areas relevant to IoT facilitate an easier adoption of smart farming and its use by farmers [19]. As farmers and their attending veterinarians, nutritionists and advisors become increasingly aware of the benefits of ICT, it will hopefully motivate them to upload data to central repositories on, for example, disease incidence, the number of live-born piglets in individual sows, feed intake and weather variables. The collection of animal-based data is advancing rapidly, with behavior data alerting farmers to health and productivity problems, as well as the physiological status of animals, such as when they are in estrus. Although the fundamental value of such data has been known for several decades (e.g., [20]), the miniaturization of recording systems has only recently made widespread use possible. For example, early versions of pedometers for dairy cows uploaded data to a computer attached to the cow’s back [21]. Still, it took several decades before pedometers were small enough to be feasible for mainstream use in dairy herds.

Processed data may collectively benefit animal production as patterns emerge or individually as perturbations in the individual animal or group of animals are detected in response to environmental variables. Furthermore, some of the world’s largest agricultural producers are promoting the use of IoT in smart farming by creating incentive programs and public policies to fund research and training [22]. Several recent reviews have been published on IoT solutions for smart agriculture, suggesting that this research field is constantly receiving new contributions and improvements [23]. Technologies used for communication and data collection solutions are presented in [24], as well as several cloud-based platforms used for IoT solutions for smart farming. An IoT architecture with three layers (perception, network and application) was enforced to analyze the application of sensor and actuator devices and communication technologies within several farming domains, such as agriculture, food consumption and livestock farming [25].

On the other hand, it was suggested that European farmers lack the knowledge to understand the benefits of ICT-based PLF [26]. The acceptance of (new) IT technologies, such as big data, computer vision, artificial intelligence, blockchain and fuzzy logic in the smart agriculture field were evaluated in [27]. A study of the consumer perceptions of PLF technologies showed that consumers expect that PLF technologies will enhance the health and welfare of farm animals while generating environmental improvements and increasing transparency in livestock farming [28]. The researchers, however, also expressed the fear that PLF technologies will lead to more industrialization in livestock farming, that PLF technologies and data are vulnerable to misuse and cyber-crime and that PLF information may be inadequately communicated to consumers. Public opposition to the industrialization of livestock production is encouraging de-intensification by farmers, either to meet government standards or to capture higher product prices. However, less intensive livestock farming utilizes more land, a commodity in short supply with a growing world population and competition from carbon farming to offset increased emissions. Recently, a book published by Wageningen Academic Press detailed the on-farm experiences (both positive and negative) of 90 authors from 16 different countries: all users, developers and academics working in the PLF field [29].

To ensure that agriculture supplies secure and nutritious food while minimizing environmental threats, farmers need specific economic incentives, help with incorporating innovation into their enterprise and knowledge exchange to encourage the use of advanced and smart technologies. Coherent agricultural, environmental, trade and R&D policies must be presented by the government. It is also vital to base policy decisions on robust, well-established scientific criteria so that the decisions are justified and can be explained to all stakeholders. The EU has been fostering PLF through funding and investment since the FP7 program. The EU CORDIS service (cordis.europa.eu, accessed on 31 May 2023) provides details on 77 forefront projects dealing with animal production systems and animal health, which have received an EU contribution of € 508 million under Horizon 2020 and Horizon Europe programs [3,30]. In light of the ongoing significant European investments in animal research and trends in the EU’s food chain [31], we recognize the necessity of a state-of-art review describing the latest technological developments (last decade) in the poultry and pig sectors, all in context and based on the actual funded and operative European projects as published in the CORDIS service. Previous studies [12] indicated that farmers initially have concerns about the usefulness of PLF tools and typically do not fully exploit them. However, this can positively change when implementing extension/education processes [32]. This literature review aims to identify how digital technologies are implemented in European livestock farms by (i) presenting a review of the state-of-the-art adoption in (1) pig and (2) poultry farms and (ii) reviewing farmers’ attitude toward and concerns surrounding the acceptance of PLF technologies, based mainly on past EU projects.

## 2. Technologies in Livestock Farming

Generally, sensors such as thermal imagery, microphones, GPS and others are used in PLF to collect real-time data [19]. Due to the significant amount of raw data collected, algorithms are often applied to aid analysis. The data can either be directly processed or immediately relayed to the farmer, or it can be transferred to the server of a service provider company where it is analyzed, and the feedback is sent to the farmer. ICT can promote learning, which in turn can facilitate technology adoption among farmers, and it has the potential to revolutionize early warning systems through better quality data and data analysis. However, the information relayed by ICT should be properly targeted and relevant if it is to affect farmers’ production decisions. The evidence [33,34] suggests that content quality and relevance are crucial. Building up human capacity, as well as the infrastructure needed to facilitate better connectivity, is also critical. In this way, the use of contact time between humans and livestock can be more productive, but it should aid good stockpersonship rather than being a replacement.

The manner by which information is delivered is also a crucial determinant of effectiveness. ICT encompasses many different technologies, from computers and the Internet to radio and television to mobile phones. Their impact varies widely depending on which specific technology is used but also on farmers’ level of technological literacy. A growing body of evidence suggests that in many circumstances, mobile phones can increase access to both information and capacity-building opportunities for rural populations in developing countries [35]. Farmers can get access to timely and high-quality information on products and inputs, as well as on environmental and market conditions. Short message services (SMS), voice messages, short video trainings, audio messages, social media interventions and virtual extension platforms that can improve peer networks (through online platforms/websites) can effectively enable farmer-to-farmer and farmer-to-experts information sharing. Audio- or voice-based question-and-answer services may overcome the limitations of text-based platforms. SMS messages can be effective for sharing simple price or weather information, but to facilitate and revolutionize learning and make knowledge widely accessible, especially in the context of adapting agriculture to climate change, other methods and modes will be necessary.

Within the framework of the AutoPlayPig project [36], funded by the EU’s Horizon 2020 program under the Marie Skłodowska-Curie grant, a comprehensive review was published on information technologies (ITs) developed for welfare monitoring within the pig production chain, evaluating the ITs developmental stage and how these ITs can be related to the Welfare Quality^®^ (WQ) assessment protocol [37]. Of the 101 publications included in the systematic literature analysis, 49% used camera technology, 18% used microphones and 15% animal attached sensors, including accelerometers and radio frequency identification (RFID) tags. The sensor technology used to measure environmental biomarkers included thermometers, an Environmental Monitoring Kit, an anemometer, an air-speed transmitter and a weather station. Most publications investigated feature variables on individual or pen levels of behavioral animal biomarkers. Most publications investigated ITs for welfare monitoring in growing pigs and lactating sows, whereas almost no publications investigated pigs during transport or sows in the insemination unit. Nearly all (97%) publications investigated welfare issues in real-time; however, only 23% properly validated their results. An analogous systematic review was published on validated PLF technologies for pig production in the context of animal welfare [38], within the framework of the ongoing EU project ClearFarm [39], funded by the EU’s Horizon 2020. Eighty-three technologies with a potential link to animal-based pig welfare assessment were found, based on 10 different types of sensors (in descending order of frequency of use): camera, load-cells (with and without RFID), accelerometer, microphone, thermal camera, photoelectric sensors, flow meter, RFID and non-contact body-temperature sensors. Of these technologies, 39% was used for fattening pigs, 33% for sows and 28% for piglets and weaned piglets. Monitored indicators included activity and posture-related behavior, feeding and drinking behavior, physical condition, health-related traits and other behaviors.

In a review of PLF in the poultry sector [40], as part of the completed EU-funded ERA-NET project ANIHWA [41], a similar segmentation was demonstrated. Fifty-two percent of the 264 reviewed publications described sensor technology, 42% described the use of cameras and only 14% described the use of microphones. Animal health and welfare constituted the most popular field of study (64%), followed by production (51%) and, by a large margin from third place, sustainability (only 8%). Most measurements used to evaluate animal health and welfare were behavior-based, with 44% of publications using locomotory behavior, followed by bird sounds (21%). Out of the 264 reviewed publications, a mere 4% described commercially available systems.

All of the data generated by the aforementioned sources need to be exploited to validate and further develop useful algorithms; however, this requires the availability of advanced infrastructure [42]. As such, big data generated from technological sources require advanced analytics for effective exploitation. Advanced infrastructure is also needed for the timely and efficient execution of these big-data-enabled algorithms prior to delivery to the farmer. The recently completed EU project CYBELE [43], funded under the Horizon 2020 Programme, aimed to introduce to all stakeholders along the agri-food value chain an ambitious and holistic large-scale High-Performance Computing (HPC) platform, offering services in data discovery, processing, combination and visualization and solving computationally-intensive challenges requiring very high computing power and capable of actually generating value and extracting insights from the data [42].

The future may enable ICT to bring even greater improvements in animal welfare and productivity. A machine learning framework to predict the next month’s daily milk yield, milk composition and milking frequency of cows in a robotic dairy farm has been developed [44]. The self-selection of rewards can contribute to animals’ freedom of choice using digital technology such as touchscreen monitors, which have already pioneered for animals in zoos [45]. The selection of foods from a variety of possible plants on offer has evolved, and allowing animals to choose resources via smart devices may improve their welfare. In rodents, enrichment leads to greater exploratory behavior and better coping with stressful conditions [46]. In pigeons, free choice is preferred to forced selection [47,48], and comparable benefits may be demonstrable in poultry and other farm animals. Primates have been most often demonstrated benefits from mastery over their environment [49], but reliable testing for livestock is yet to be undertaken.

## 3. Scientific and Commercial Review of Operational Concepts and Technological Solutions in the Pig Sector

Various areas of research are reflected in the European studies. Among them, several areas are prominent.

Weighing optimization–The completed European project ALL-SMART-PIGS [50], funded by Horizon FP7, was one of the first EU projects to showcase commercialization as a main focus. The Weight-Detect^TM^ application (PLF Agritech, Toowoomba, Australia) is an innovative video image analysis system that determines the group average weight of a pen of animals by a video observation system. It enables farmers to determine growth and any weight-based indexes without physically weighing the animals [51]. Pig weighing optimization was selected for the evaluation and technical validation of a platform in the aforementioned CYBELE project [43]. The tool is a convolutional neural network that takes images and captures videos above the pens of fattening pigs throughout their weight gain and encodes these images into a latent vector representation. Together with additional relevant information, it estimates the mean ± SD live weight of the pigs in the pen. Body weight recording was the subject of the ClearFarm project [39]. The automated estimation of body weight was conducted by a depth camera (iDOL65, dol-sensors a/s, Aarhus, Denmark) [52] placed above the individual feeding station or three-partitioned feeder, which worked in combination with an RFID system installed in the feeding stations. The performance of the depth camera and its underlying algorithm was satisfactory at both installations; however, a lack of frequent maintenance, changes in pens’ uniformity and dietary shifts may compromise image sampling and body weight estimation. Similar results were reported by [53].

Play behavior–In general, the scientific literature supports the use of play behavior as an indicator of good animal welfare and affective states that are valanced [38,54,55,56]. The AutoPlayPig project [36] aimed at taking the first steps in developing a system for automatic detection of play behavior in young pigs as an indicator for welfare assessment. This was accomplished by developing an algorithm to extract heart rate (in beats per minute) from raw video data of an anesthetized and resting pig wearing an electrocardiography (ECG) monitoring system, thus combining ethology and computer science into one field of Computational Ethology (CE) [57]. Play behavior frequency over the process of weaning piglets was investigated in the ClearFarm project [39] by analyzing the effects of two weaning methods [conventional weaning: two litters mixed in a weaner pen of different size and design vs. litter staying in the farrowing pen after removing the sow] and two genetic hybrids [DanBred Yorkshire × Landrace vs. Topigs Norsvin TN70 Yorkshire × Landrace] [58]. The results showed that weaning stress in pigs may be reduced both by using a genetic hybrid pig breed with higher birth and weaning weights and by keeping litters intact in a familiar environment after weaning. A first attempt at the automatic detection of locomotor play behavior in young pigs from video by classifying locomotor play from other solitary behaviors, including standing, walking and running, is presented in [59]. Two methods were tested: a method utilizing the Gaussian Mixture Model (GMM) for quantification of movement combined with standard machine learning classifiers and a method utilizing a deep learning classifier (CNN-LSTM) on the raw segmented video. The deep learning classifier obtained higher Recall, Precision and Specificity values.

Tail biting–In intensive piggeries, tail biting is common and is considered an indicator of negative welfare [59]. This issue is addressed in the on-going European project Code Re-farm [60,61] using Duroc × (Landrace × Yorkshire) piglets in free-farrowing pens. In conclusion, the study’s proposed method detected tail-biting behavior from video sequences of entire pig pens, claiming its CNN-LSTM model to be superior to the CNN-CNN model. Moreover, the study found that implementing principal component analyses on the extracted spatial feature vectors can increase the performance compared to using all extracted features.

Virus detection–A novel and affordable field diagnostic device, based on advanced, proven, bio-sensing technologies to tackle six important swine diseases has recently been developed within the Horizon 2020 SWINOSTICS project [62]. The diagnostic device allows threat assessment at the farm level, with the analytical quality of commercial laboratories. The SWINOSTICS mobile device can simultaneously analyze four samples to detect six of the most important swine viral pathogens: Porcine Parvovirus (PPV), Porcine Circovirus 2 (PCV−2), Classical Swine Fever Virus (CSFV), Porcine Respiratory and Reproductive Syndrome (PRRSV), Swine Influenza Virus (SIV) and African Swine Fever Virus (ASFV) [63,64,65]. According to [63], the SWINOSTICS device can be used at the farm level to assess the health status of newly purchased animals and to identify PPV- and PCV−2-infected animals before the onset of clinical disease, thus supporting evidence-based disease control strategies.

Additional research areas–Two smart farming applications ready for commercialization on European pig farms were evaluated within the ALL-SMART-PIGS project [50]: a feed intake measurement device (Feed-Detect^TM^, PLF Agritech, Toowoomba, Australia) and an environmental monitoring (Enviro-Detect^TM^, PLF Agritech, Toowoomba, Australia) device [18,66,67]. In addition, a sound monitoring device (originally developed within the Catholic University of Leuven) was also evaluated to facilitate early detection of respiratory diseases on pig farms [51]. The sensor outputs of these technologies have been combined in FarmManager management system (schauer-agrotronic.com, accessed on 30 June 2023). Chain feed optimization was realized by using traceability in an online digital logbook, including a SMS-based warning system for the farmer [50]. Technical and technological issues and their adequate implemented solutions, the technological impact of installed PLF and the business impact of their usage were all considered. Sustainable pig production was another demonstrator selected in the CYBELE platform [43]. It utilized data from different barn sensors to monitor individual pigs’ feeding and drinking behavior on a continuous basis. Based on multivariate algorithms, problems at the individual and group levels could be detected. The tool exhibited an improvement in the average health prediction precision and sensitivity in warning systems for pigs compared to a previous model using the same dataset. In Table 1, we present some of the technological advancements and scientific research involved with PLF in the pig sector. As evident from Table 1, the volume of research in the pig industry concludes 42 projects and 79 peer-reviewed articles.

## 4. Scientific and Commercial Review of Operational Concepts and Technological Solutions Used in the Poultry Sector

PLF development in the EU has most commonly focused on broiler farming, followed by laying hens. Modern broiler strains in intensive production systems reach their target weight in just 5–6 weeks or less [141]. This short life span means that it is difficult to maintain a balance between production objectives and bird welfare. A review of the trends in PLF in the broiler production industry, supported by the Irish Innovation Partnership Pathway, [142] elaborated that while the use of electro-chemical sensors in precision farming is quite common, the use of state sensors measuring physical properties such as temperature, acceleration or location is still at a preliminary state of deployment. As the cost of wearable sensors decreases, the option of fitting a large number of birds with these physical state sensors seems more and more feasible. However, a recent study in Flanders, Belgium [143], showed that the broiler chickens’ behavior was substantially interrupted after the wearable sensors were fitted. Within the remote sensors technologies, the Near Infrared (NIR) sensors may provide advanced data such as the thermal profiles and physical properties of the chickens, as well as measurements of CO_2_. Non-point sensor datasets, mainly video and still image datasets for continuous monitoring, have been implemented for monitoring bird performance and estimating average bird weight [142].

The opportunity exists in the poultry sector to monitor ammonia concentrations using multiple sensors feeding data into a central console. However, developing an adequate ammonia sensor is still a challenge that has not been resolved in a satisfactory way. One of the main outputs of the European project EU-PLF [144], funded by Horizon FP7, was an embryonic blueprint for commercializing PLF type technologies. Within EU-PLF, broiler activity was defined as a key indicator for welfare and health. The remote camera detection of broiler behavior enabled the development of an early warning system to alert managers to unexpected broiler behavior with 95% true positive events [145]. Relationships between leg problems, such as Foot Pad Dermatitis, and environmental variables (i.e., temperature humidity index, THI) were detected, which aided in developing an automated prediction system [146,147]. An analysis of behavioral responses, avoidance distances and gait scores, to human (farmer) presence yielded an indicator for broilers’ fear of humans. Finally, indoor particulate matter concentrations (dust) detected by sensors showed a strong correlation between emissions and bird activity [148].

Alerting farmers to welfare problems in real-time, especially during winter nights when ventilation is low, allows for fast and targeted interventions, which will immediately benefit the flock compared to traditional welfare assessments that have usually occurred on the next morning [149]. Ammonia concentrations are often higher during the daytime in livestock buildings due to increased evaporation via higher temperatures, greater movements of birds and increased airflow [150].

Research and development in poultry disease identification and control should be prospective and incorporate new technologies and should pay special attention to zoonotic diseases. Such a perspective was demonstrated by [151], in the framework of two completed Horizon 2020 projects–SMARTDIAGNOS [152] and VIVALDI [153]–with the study of two *Campylobacter* species–*C. jejuni* and *C. coli*–in poultry flocks. These two species account for most human campylobacteriosis [154], and poultry and poultry products are considered to be the main sources of disease transmission [155]. To tackle this, a simple and rapid Loop-Mediated Isothermal Amplification (LAMP) assay was used to detect *C. jejuni* and *C. coli* in chicken feces.

Broiler production systems must be optimized to enhance their energy/resource efficiency, minimize carbon footprint and create sustainable supply chains by developing the necessary infrastructure across all stages of production, including breeding, hatching, rearing, processing and distribution to consumers. Collaborative research and advanced technologies can help tie together the different components of the system and their relationships. The consequences of not supporting farmers in implementing new technologies may result in the loss of social licence and even threaten the poultry industry’s premier position in the global marketplace and the ability of the industry to provide safe and nutritious poultry products to consumers worldwide [156]. The lack of collaboration between the private and public sectors and the lack of innovative ways to articulate concerns from producers and consumers to policymakers remain barriers to technological adoption [13]. In Table 2, we present some of the technological advancements and scientific research involved with PLF in the poultry sector. As evident from Table 2, the volume of research in the poultry industry is more than twofold lower than that of the pig industry, featuring 16 projects and 27 peer-reviewed articles.

## 5. Scientific Review of Farmer’s Attitudes and Obstacles in Acceptance of Technological Solutions in the Livestock Sector

Qualitative and quantitative assessments of the attitudes and barriers to PLF technology adoption have shown the manifold factors that influence farmers’ technology decisions and highlighted the economic, socio-demographic, ethical, legal, technological and institutional aspects that need to be considered for widespread technology acceptance [178,179,180]. They also showed that “innovation uncertainty” has led to a rather slow uptake of precision technologies by farmers thus far [181]. Kling-Eveillard et al. [182] mentioned that farmers using PLF depict a stockpersonship that has not fundamentally changed but which involves new components such as tasks, skills and schedules. Moreover, we know that the manner of PLF usage varies between farmers, i.e., the degree to which the farmer delegates tasks to the equipment [183]. Noting this, specific attention should be given to the study area, as the attitudes and barriers to technology adoption may vary depending on the socio-economic and cultural situation of the region. Studies in countries with strong educational institutions and high standards of living may experience different barriers than farmers in low-income countries that lack more basic needs for technology adoption (such as internet access, education, monetary funds, social capital, etc.). This review section focuses primarily on European and American study sites and gives an overview of prominent farmers’ perspectives relevant for technology adoption.

The most reported aspect in almost any region is the fear of high investment costs that are needed to enable PLF [180,184,185,186,187,188,189]. This barrier is particularly prominent for smaller production sites, as the expected investment returns are more limited compared to big farms [185,188,190].

Aack of trust in the technological capabilities and robustness of the technology was another frequently reported factor that affected technology adoption [179,180,189,191]. As trust has many different notions, there are several associated aspects that directly or indirectly influence the confidence in specific technologies. Farmers that are in close proximity to other farms and are part of a wider network tend to adopt novel technologies quicker and more optimistically [180]. Trust in technology is higher if colleagues have used the technology effectively before [192]. This networking effect was also shown by [184], who found that 68% of farmers make adoption decisions based on information obtained from colleagues. Trust may also be a relevant factor in the sense of security and privacy. As modern PLF technologies are embedded in an IoT environment and are often accompanied with decentralized data storage (e.g., cloud or edge devices), concerns about data safety may arise. Privacy and security concerns are one of the most prominent barriers that inhibit digital technology adoption by farmers in Wisconsin [179]. However, a study by [180] found that the participants did not express any concerns about data privacy issues but are optimistic about the positive influence of collective data processing.

Another important factor influencing the purchasing decision in PLF technologies is the perceived usability of PLF products. This includes closely related factors such as the complexity of technologies, the necessary education to install, interpret and use of the technology as well as the external dependencies on service providers and vendors that are associated with it [180,185,188]. Scholars [14] and farmers [189] have both highlighted that the necessary technology integration and interoperability among PLF relevant systems further hinder usability and therefore harden existing barriers.

Some other factors, such as technological relevance or lack of awareness have been identified by individual studies [179,180,189]. These are believed to be of minor importance, and in practice, most attitudes are closely linked to the already mentioned farmers’ characteristics and associated barriers. This also highlights the potential of positive side-effects if one addresses the individual fears and needs of farmers in terms of technology adoption.

Interviews and surveys constitute the main methodology through which the voices of farmers are heard, but it is always important to consider their accuracy, especially in relation to sensitive animal welfare concerns. In the EU-PLF project [144], farmers voiced their hesitation to purchase PLF technologies, unless its derived benefits are clear and unequivocal, and they also had concerns about maintenance. The issues and importance of training on-site, providing professional on-demand and continuous support, especially concerning animal welfare assessment [193], and establishing demonstration farms were stressed. Almost all farmers were afraid of losing direct contact with the animal–their “care relationship” (particularly pig farmers rather than poultry farmers). Prospected environmental and welfare regulations hindered the farmers from investing due to uncertainty about whether future market conditions will allow their investment to be repaid. Further negative associations with PLF were its perceived complex operation and a partial ability (at best) of the farmer to understand the information in a simple and coherent way and act upon it. As one farmer stated: “The data doesn’t have to be 100% accurate, but 100% reliable” [193].

Farmers from the pig and poultry sectors in the UK and Spain interviewed in the EU project Feed-a-Gene [194] stressed the importance of providing accurate and complete information to farmers and the need for a detailed evaluation of novel technologies in a commercial setting before more widespread adoption. Interviewees from the pig sector favored the concept of precision feeding and the resultant improvement in feed conversion efficiency and improved animal welfare; however, farmers from the poultry sector (in Spain) were largely unenthusiastic. In the pig sector, the benefits seemed clearer for gestating sows than for breeding pigs.

The expected high costs for investment have led to scepticism about whether gains in feed efficiency necessary to justify the investment would be realized. This would particularly apply if existing buildings, infrastructure and feeding systems could not be simply adapted, as most farmers believed to be the case. Concerns were raised about the necessary skill level of operating such precision feeding systems, as it would require skilled labor, which is expensive and could increase labor costs. Additionally, equipment suppliers must be able to provide a fast and reliable on-farm repair service, which requires enough skilled staff.

One of the targets of the SusPigSys European initiative [195] (part of the ERA-Net Cofund activity SUSAN) is to promote farmers’ wellbeing. Farmers of pig production systems in seven EU member states (Austria, Germany, Finland, Italy, Poland, the Netherlands and the United Kingdom) participated in national workshops with stakeholders, where the important social implications for farmers themselves were discussed. The participants from Germany, Finland and the UK stressed the importance of consumers’ power along the supply chain and societal acceptance of the public image of pig production and the farming profession, highlighting the disconnect between the industry and the consumers. Most EU pig farmers are concerned with animal welfare and environmental impacts, as well as the economic survival of their businesses. Farmers also would like people outside the industry to better understand the demanding work of pig farmers in producing food sustainably. This latter point is echoed in the completed FP7 project PROHEALTH [196], where [197] examined the attitudes of the public in five EU countries (Finland, Germany, Poland, Spain and the UK) toward intensive animal production systems and production diseases in the broilers, layers and pig sectors. Most alarming is that a significant portion of the public is not familiar with modern animal production; nonetheless, they perceive intensive production systems negatively, which subsequently influences their consumer behavior.

To counter this and other negative associations with PLF technologies, the “LivestockSense” project [198] was implemented in seven different EU countries to encourage PLF technology adoption and increase the general understanding of these technologies. An online quantitative survey was undertaken, and follow-up interviews, as well as focus group discussions (FGDs), were organized to obtain qualitative results. The quantitative questionnaire results demonstrated that the existing level of automatization on the farms, the average age of the livestock buildings (and associated production technologies) and the availability of internet connectivity were clear indicators of livestock producers’ “readiness levels” to adopt PLF technologies. In the second half of the project, complex software development was undertaken to create an integrated cloud-based ICT tool that captured the key outcomes of this project, including the (1) user classifier, (2) the benefit calculator and (3) advice generator software applications.

The possible advantages mentioned by interviewees included the possibility of working with large animal groups (however, this is only possible for big farms), gains in space by removing passageways and walls, retaining young people in rural areas, as they might find careers in pig husbandry more attractive, and a reduction in the environmental footprint of pig farming. A Swedish farmer in the egg production sector who was interviewed as part of the completed EU project SURE-Farm [199] advocated that new machines and robots have helped to eliminate heavy physical work, which previously limited the opportunities for older farmers and farm workers to continue working [200]. The Horizon Europe Thematic network BroilerNet [201], with 25 partners in 13 countries, is aiming to create 12 innovation networks at national level and three EU level networks of broiler farmers, advisors, supply chain integrator companies, farmers’ organisations, researchers, and veterinarians. The project will focus on environmental sustainability, animal welfare and animal health management. By identifying the most urgent needs of broiler farmers, the network will collect and evaluate good practices that are able to meet these needs.

Nevertheless, it is important for future research in both sectors to focus not only on the technological improvements of tools and sensors but also on the aspects of environmental, economic and social sustainability of livestock production that impact both farmers and the community and consumers [202].

## 6. Conclusions

PLF systems can help to increase production efficiency and meet consumer expectations of more environmentally and welfare-friendly production at a time when there is extreme pressure on land availability for food production that deters farmers from reducing the intensity of their operations. Several EU-funded projects have helped to identify and develop PLF technologies that could benefit the livestock industries, particularly in the pig and poultry sectors. The large volume of research in the pig industry is welcomed, but attention must be paid to the global and European trends of an increase in broiler-meat production, which is growing faster than any other meat type, including pork production. Therefore, greater research efforts are required in the poultry industry, with particular recommendation for the development of enhanced research infrastructures in the sectors of laying hens, turkeys and quails, etc., on the basis of their being underrepresented in the plethora of active research projects in Europe.

The EU continues to lead in this field, although it is expected that other regions, such as PR China and the USA, could be important in scaling up the production of PLF technology once its value is proven.

Considerable obstacles to widespread PLF adoption have been identified and must be addressed. High investment costs, a lack of trust in the technology, uncertainty in the future market for their products and the usability of the technologies have all been identified as impediments to adoption. Fifty-eight percent of European farm managers are 55≤ years of age, and of them 33%, are over 65 years. Most of them work on small-size farms, mainly family farms, which constituted a staggering 94.8% of EU farms in 2020. These data illustrate the magnitude of the challenge in embedding and implementing PLF in contemporary livestock agriculture in Europe. It can be concluded that there is enormous importance in the integration and involvement of stakeholders from the fields of social sciences in order to mediate the farmer–technology interface. Given the growing frequency of crises, such as the recent COVID-19 epidemic, it is imperative to supply farmers with adequate platforms for installation, service-oriented accompanying and a significant financial support network that will allow them to more realistically and competently deal with prevailing barriers in terms of investment and innovation uncertainty. In particular, the investigation of farmers’ attitudes and obstacles to acceptance of technological solutions in the livestock sector should be integrally coupled with any technological development.

We perceive the increased use and uptake of PLF in the future as imperative. The future management of issues concerning the unknown costs and benefits of PLF systems should be the responsibility of stakeholders. This need is not only urgent for farmers but also important for financial institutions and governments offering support and subsidies. To eliminate distrust in technology and provide solutions that meet farmers’ needs and are well-suited for farming conditions, developers must act. Novel devices, sensors and technologies that have a clear and quantifiable advantages for farmers of the pig and broiler sectors should be demonstrated, targeting environmental monitoring and the AI-driven analysis of livestock behavior in order to maximize the economic production and the environmental and welfare performance of the animals. To effectively implement PLF adoption, it is crucial that farmers are clearly informed of the minimum farm infrastructural requirements. Additionally, to address any concerns related to using advanced technologies, enhanced collaboration between farmers, scientists and engineers is required, coupled with targeted education, training and information-sharing.

## Figures and Tables

**Table 1 animals-13-02868-t001:** Examples of funding programmes and agencies using PLF technologies in the pig sector developed within the last decade in the framework of European-funded projects. Studies without clear acknowledgment of the funding source were not included.

Pig Sector		
Funding Programs and Agencies	Project Name	Output [Reference]
German Federal Ministry of Food, Agriculture and Consumer Protection (BMELV) via the Federal Office for Agriculture and Food (BLE) in the framework of the innovation support program		Automatic detection; body core temperature; infrared thermography (IRT) measurements [68]
Marie Curie BioBusiness FP7–PE–PLEITN–2009–2014		Automatic detection; aggressive behavior, by: activity index; multilayer feed forward neural network [69]; camera recording [70]
FP7–Seventh Framework Programme	ANIMA–CHANGE	Decision support tool; feeding strategies; fattening pigs; lactating sows; climate change [71]
German Research Community (DFG)	HE 6419/1–1	IRT; thorax; lung alterations [72]
FP7–Seventh Framework Programme	ALL–SMART–PIGS	Weight-detect; video image analysis; group average weight [45]; feed sensor; weight of feed [45]; cough monitor; microphones; sound talks online application [45]; camera system; activity and the distribution of pigs; occupation-density index; activity index [45]; air quality sensor; airborne pollutants [45]; farm traceability system; production data; health monitoring; transportation [45]
FP7–Seventh Framework Programme	EU–PLF	Pig cough identification; health monitoring [73]; camera recording; animal shapes; animals’ position and movement [74]
Flemish Government Agency for Innovation by Science and Technology (IWT), Belgium	Project number: 080530/LBO	Automatic monitoring; pig weights; image analysis [75]; behavioral activity; human labeling; automated video analysis [76]; automated scream detection; sound recordings [77]
Douglas Bomford Trust, UK		Microsoft Kinect sensor; normal walking in pigs [78]
Partly funded by Danish Meat Research Institute	Monitoring Animal Wellbeing	Optical flow (OF); monitor pig herd movement; slaughterhouse [79,80]
The French Agency for Ecological Transition-ADEME		Feeder prototype; precision feeding; RFID ear tag [81]
The Swedish Research Council for Environment, Agricultural Sciences and Spatial Planning (FORMAS)		Image analysis; proportion of pigs [82]
Provincial Government of Niederösterreich, Austria	PIGwatch	Accelerometer; detection and classification of: nest-building behavior; sows [83]; sow’s postures [84]; camera monitoring; farrowing process; sows [85]
Federal Office for Agriculture and Food, Bonn, Germany (BMEL) under the innovation support program (FKZ 28154T0910)	Electronic Animal Identification Systems Based on Ultra High Radio Frequency Identification	Light barrier sensors; predicting and detecting; parturition onset [86]; high radio frequency (UHF-RFID); reader; antennas; passive transponder ear tags; monitoring visits; growing-finishing pigs; liquid feeding [87]
Finnish Ministry of Agriculture and Forestry	SowMonitor	Detection; farrowing; sows; wireless 3D accelerometers; activity [88]
Innovate UK	Green Pigs	Image processing; detection; group lying behavior; mounting behaviors; position changes; pens enriched; [89,90,91,92]
ICT–AGRI ERA–net; German Federal Office for Agriculture and Food (BLE); Agency for Innovation by Science of Technology (IWT Flanders)	PIGWISE Project SB 111447	HF RFID; identification: feeding profiles [93]; drinking behavior [94]; HF RFID [95]; feeding patterns; automated monitoring and warning system [96]
FP7–Seventh Framework Programme	PROHEALTH	Posture assessment; sows; parturition; A tri–axial accelerometer [97]; deep learning; early detection; respiratory disease; growing pigs; environmental sensor data [98]; accelerometery; lying behavior; free-farrowing sows [99]
FP7–Seventh Framework Programme; The Spanish Ministry of Economy and Competitiveness; Junta de Andalucía and the European Regional Development Fund (ERDF)	RAPIDIA-FIELD	Biosensors; accelerometer embedded in an ear tag; body temperature; motion, early detection; infectious diseases [100]; animal activity; video infection with African swine fever (ASF) [101]
IFIP–Institut du porc; INRA; Rf-track; The Chambres d’agriculture de Bretagne; French government: Ministère de l’agriculture et de la forêt		Ear tag accelerometers; sows; activity; improving feeding [102]
German Federal Ministry of Education and Research (BMBF) provided through the German Aerospace Center (DLR)	Grant 01KI1301D (MedVet-Staph 2)	Real-time location; ear tag; position large number; group-housed sows [103]
Biotechnology and Biological Sciences Research Council (BBSRC); Innovate UK (101906); Supported by Zoetis UK Limited, Innovent Technology Limited; RAFT Solutions Ltd.; Harbro Limited	Grant BB/M011364/1	Depth video camera; 3D pig positions; measure multiple behaviors [104]
Innovate UK as part of the Agri-Tech Catalyst; Biotechnology and Biological Sciences Research Council (BBSRC); Rural and Environment Science and Analytical Services Division of the Scottish Government	Early detection of tail biting in pigs using 3D video to measure tail posture (132343)	Early warning; tail biting; time-of-flight 3D cameras; pig tail posture [105]
Deutsche Forschungsgemeinschaft	DFG: KR 2024/7–1	Lameness detection; gestation sows; ear-tag sampled acceleration data [106]; ear sensor; color cameras; onset of farrowing [107]
Regional government Xunta de Galicia through the “Program of consolidation and structuring of competitive research units”	GPC2014/072	Prediction; CO_2_ concentrations; livestock building; weaned piglets; WNN model [108]; daily activity; piglets; passive IR detector (PID) [109]
Natural Environment Research Council, UK as part of the Sustainable Agriculture Research and Innovation Club in collaboration with AB Agri	Grant Reference: NE/P007945/1	Face recognition; dataset of pig faces [110]
Institute for Promotion of Innovation through Science and Technology in Flanders (IWT); Orffa; VDV Beton; Boerenbond; AVEVE; INVE; Boehringer Ingelheim	Grant number 090938	Mobile Claw Scoring Device (MCSD); cameras with light-emitting diode (LED) lights [111]
Danish Council for Strategic Research; Green Development and Demonstration Programme under the Ministry of Food, Agriculture and Fisheries, Denmark	PigIT, Grant number 11–116191 IntactTails j.nr. 34009–13−0743 StraWell j.nr. 34009–13−0736	Pen level temperature; predicting pen level outbreaks; diarrhea; pen fouling [112]; body weight monitoring; growing-finishing pigs [113,114]; multivariate spatial dynamic linear model (DLM); drinking patterns; predict outbreaks; diarrhea; pen fouling [115]; prediction; tail biting; drinking behavior; temperature of the pen [116]; prediction; pen fouling; machine learning; position of the pigs [117]
Breed4Food Program; Netherlands Organisation for Scientific Research (NWO); Dutch Ministry of Economic Affairs through TKI Agri and Food Project	Smart Animal Breeding with Advanced Machine Learning (Grant number: 14295)	Muscularity; RGB–D computer vision; machine learning [118]
Federal Ministry of Food and Agriculture (BMEL) via the Federal Office for Agriculture and Food (BLE) under the innovation support program		Radar sensors; monitor animal behavior [119]
FP7–Seventh Framework Programme	EFFORT	Electrostatic dust fall collectors (EDCs); description of resistome; bacterial microbiome; pig; farm dust [120]
Horizon 2020–Part of the ERA-Net Cofund SusAn	PigSys	Data warehouse; data storage and processing [121]; camera system; air characteristics and compositions; spatial distribution; behaviors [122]; machine vision; detect pig postures; different farming and rearing conditions [123]; life cycle assessment (LCA); evaluate pig production systems [124]
Horizon 2020	SWINOSTICS	Swine diseases; field point of care (POC); diagnostics toolbox; advanced bio-sensing; photonics technologies; emerging and endemic viruses; swine epidemics [63,64,65]
Horizon 2020	Feed–a–Gene	Autonomous localizing and tracking; RGB cameras [125]; precision feeding; growing-finishing pigs [126]
Biotechnology and Biological Sciences Research Council in the UK	Grant BB/N012518/1	OF algorithms; activity changes; tail-biting [127]
Ministry of Rural Development and Consumer Protection of Baden-Württemberg, Germany; Project LabelFit by funds of the Federal Ministry of Food and Agriculture (BMEL) via the Federal Office for Agriculture and Food (BLE) under the innovation support program	Lan–wirtschaft 4.0: Info–System	2D camera imaging; deep learning; position and posture detection [128]
Spanish Ministry of Economy within the scope of Eurostars Projects from European Union. This project is based on international cooperation led by PigCHAMP Pro Europa	Wireless Livestock Tracking system (WILT)	Activity patterns; free movement livestock; animal located sensors; individual accelerometer data [129]
UK Biotechnology and Biological Sciences Research Council	Grant BB/I005641/1	Computer vision image analysis; rapid defence cascade (DC) behavior [130]
Federal Ministry of Food and Agriculture (BMEL) via the Federal Office for Agriculture and Food (BLE) under the innovation support program (FKZ 2817902615)		PID; measuring group activity; fattening pigs [131]
Austrian COMET–K1 competence centre FFoQSI	Grant number 854182	Deep learning; tail-biting; IP camera; infrared spotlights [132]; ear tag; acceleration; indication; sows; confined in crates [133]
Horizon 2020	ClearFarm	Body weight; depth camera [52]; play behavior; weaning; video recordings [58]; locomotor play behavior; young pigs; video [134]
Horizon 2020	Code Re-farm	Deep learning; pen-level estimation; slaughter pig; weight distribution [135]; video-based classification; tail-biting behavior [61]
Horizon 2020; Green Development and Demonstration Programme under the Ministry of Food, Agriculture and Fisheries, Denmark Biotechnology and Biological Sciences Research Council, UK; Zoetis Inc.; Harbro Nutrition Ltd.; Innovent UK Ltd.; RAFT solutions Ltd.	HealthyLivestock IntactTails j. nr. 34,009–13–0743 Agritech Research grant (BB/M011364/1)	Camera-based data of: feeding behavior [136]; postures and drinking behavior [137]; tail-biting behavior [138], deep learning models; water and feed consumption; weight; weaned piglets; connected feeders, connected drinkers, automatic weighing stations, RFID ear tags; early detection; diarrhea [139]; behaviors of a group; video data [140]

**Table 2 animals-13-02868-t002:** Examples of funding programs and agencies using PLF technologies in the poultry sector developed within the last decade in the framework of European funded projects. Studies without clear acknowledgment of the funding source were not included.

Poultry Sector		
Funded Under	Project Name	Output [Reference]
FP7–Seventh Framework Programme	EU–PLF	Real-time modeling; indoor particulate matter concentration; broiler activity; ventilation rate [145]; sound recording; vocalization analysis; real-time analysis; broilers: internal pipping stage (IP) [71]; growth rate [157]; model weight [158]; real-time camera; activity and distribution; broiler flocks; visual welfare scores [146]; camera; gait score; lameness; broiler flocks [159]; volatile organic compounds; recognition of enteric pathologies [160]; prediction model; gait score; broilers; flock distribution; bird activity levels; body mass [161]
Dutch Ministry of Economic Affairs		Welfare Quality^®^ assessment protocol for broiler chicken welfare [162]
Green Development and Demonstration Program, established under the Ministry of Food, Agriculture, and Fisheries, Denmark		LED color temperature; preference; behavior; welfare; performance [163]
FP7–Seventh Framework Programme	WELFARE INDICATORS	Animal-based welfare indicators (AWIN); protocol; Turkey [164]
UK Biotechnology and Biological Sciences Research Council (BBSRC)	Grant no. BB/K001388/1	OF analysis; detection; chicken flock’s infection; *Campylobacter* [165]; camera monitoring; movements; broiler flocks; prediction; footpad dermatitis; hockburn [166]
FP7–Seventh Framework Programme	EFFORT	Electrostatic dust fall collectors (EDCs); description of resistome; bacterial microbiome; poultry farm dust [120]
Horizon 2020	Hennovation	E–learning course; knowledge innovation networks [167]
UK Biotechnology and Biological Sciences Research Council (BBSRC)		Behavioral; physiological response; laying hens; backpacks; thermography, direct observations; weighing [168]
Horizon 2020	Feed-a-Gene	Processing methods; European soybeans; novel ingredients; microalgae; macroalgae, duckweed, yeast protein concentrate; bacterial protein meal; leaf protein; insects; future chicken diet formulations [169,170]
KU Leuven internal research grant		Sound (chicken sneezing); monitoring; poultry health [171]
Horizon 2020; GUDP - Green Development and Demonstration Programme, Denmark	SMARTDIAGNOS VIVALDI ISOAMKIT No. 34009–15−1038	Rapid detection; *Campylobacter* spp.; broiler production chains; optimized a loop-mediated isothermal amplification (LAMP) assay [151]
Dutch Ministry of Economic Affairs and the Breed4Food partners Cobb Europe, CRV, Hendrix Genetics and Topigs Norsvin	TKI Agri and Food project 16022	Ultra-wideband (UWB) tracking; individual activity; group-housed broilers [172]
Swedish Farmers’ Foundation for Agricultural Research and SLU Ekoforsk	Project number H1343143	Welfare Quality^®^ protocol; lameness; contact dermatitis; cleanliness; thermal comfort; litter quality; human–animal relationship [82]
Horizon 2020; Horizon 2020 ERA-Net; UK Biotechnology and Biological Sciences Research Council (BBSRC)	VetBioNet Delta-Flu ANIHWA MICHIC project ANR−14-ANWA−0001 Grant BB/M028208/1; Grant BB/N023803/1	Precision-cut lung slices (PCLS); immunologically mature conventional and specific-pathogen free (SPF) chickens [173]; cameras; OF; flocks of chickens; behavior; welfare; group level [174]
Biotechnology and Biological Sciences Research Council Doctoral Training Programme Award	Grant no BB/J014508/1	Supplementary UVA and UVB wavelengths; performance indicators; welfare indicators; [175,176]
Horizon 2020	HealthyLivestock	Elevated platforms; enrichment tool; weight control; usage behavior; animal activity; early detection; animal welfare [177]

## Data Availability

No new data were created.

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
