# Peer review of "Farmers’ Perspectives of the Benefits and Risks in Precision Livestock Farming in the EU Pig and Poultry Sectors"

_animals, 2023, doi:10.3390/ani13182868_

Round 1

Reviewer 1 Report

Summary

This paper aims to highlight the importance of precision livestock technologies in incorporation into smart farming design systems towards improving animal welfare, production and food sustainability. It does this with a focus on the poultry and pig industries in the EU and does a good job of highlighting key innovation projects in both along with highlighting the main barriers and considerations.

General concept comments

The review topic is fairly well-rounded and complete for the specific aspects it states that it is reviewing. The only possible weakness would be in how this information can be made more relevant. Specific suggestions for future actions could facilitate more discussion in the scientific community and particularly with engagement with governments and farmers. Whilst a review of the technologies, their use, their levels of research and some of the barriers all in one place is great, having some defined suggested future requirements for increased use and uptake would benefit further.

Specific comments

Lines 19 and  87 – Arguable to call smart farming a new concept particularly as you are using a reference that is over 10 years old to justify it

Line 37 - “Precision livestock farming (PLF) refers to” refers appears to be the wrong wording for this sentence I would suggest it might offer “the continuous capacity of agriculture to contribute”

Line 98 – PLF doesn’t necessitate that data is transferred by communication networks to servers you can utilise direct transfer of data from devices and it still falls under the umbrella of PLF I would add the work usually to make this clear

115 – The consideration of gender feels like a throwaway line without expansion on the concept

Line 119-120 – Strong claim to say they will be more likely to do this, particularly without a reference to a study that suggests this is the case. You could maybe say it would make it easier for them to upload data but that doesn’t mean they are likely to. Particularly as many studies suggest farmers are very protective about the use of their data

Author Response

Dear reviewer,

We thank you for supporting our cause and acknowledging the importance of this review. We have taken your advice with regards to the general concept comments, i.e., relevancy and suggested future requirements for increased use and uptake. Thus, we added a paragraph at the end of the Conclusion to answer these issues (Lines 578-591).

As per the specific comments:

Lines 19 and  87 - the word "new" was discarded.

Line 37 - we thank you for your suggestion. We replaced "refers to” with "offer".

Line 98 – we thank you for your suggestion. We added the word "usually" to clear the concept of data transfer.

Line 115 - we thank you for your insight. We followed with deleting gender reference in line 115.

Line 119-120 - we thank you for your remark. We moderated the sentence and reworded it to: "As farmers and their attending veterinarians, nutritionists and advisors become increasingly aware of the benefits of ICT, it will hopefully motivate them to upload data to central....". (Currently lines 117-120)

Reviewer 2 Report

Overall, this is a really good review about current progress in precision livestock farming technologies and the thoughts and discussions related to it, which has great significance in inspiring researchers and policy makers who work in this field.

The article was well written with sufficient references, especially covering recent research projects mostly funded in EU focusing on swine and poultry sectors. The authors also reviewed studies related to farmer feedbacks toward PLF, which is my favorite part. As more and more cutting-edge technologies have been applied in livestock and poultry farming, one thing that was always ignored is whether farmers really can be benefited from those technologies. For researchers, it feels rewarding that an advanced algorithm or technique can be transplanted or applied in the field of agriculture successfully. But will farmers have the same feeling? The authors conducted a good review revealing studies that interviewed farmers about their opinions regarding PLF technologies and animal welfare, which helps researchers re-think about the motivation of conducting relevant studies.

There are minor corrections that related to formatting or reference citations that need the authors to address. 

1. The first two sentences in abstract may be moved to introduction.

2. Line 130, 141, 142, and 144 all have citation numbers as subjects in the sentences. It should be re-written.

3. Another question is for poultry production, most of projects reviewed in the study were about broiler production. It seems like there is only one research related to egg production that was mentioned in the context (line 535-536.) How about egg production in EU under the circumstance? Will the egg production industry face the same risk like broilers? 

Author Response

Dear reviewer,

We thank you for supporting our cause and acknowledging the importance of this review. We have taken your advice with regards to the minor corrections you raise:

  1. We thank you for your remark. We percieve the first two sentences as an opening that converge the reader to the time and place discussed and with a view to the future, reinforcing the necessary role of PLF. Therefore, we decided to leave them in place.
  2. Line 137, 141, 142 and 144 have been re-written. (Currently Lines 140-146).
  3. We thank the reviewer for this important insight. Indeed, most of projects reviewed in the study were about broiler production. Two research projects are related to behavioural response of Laying hens [168] and ‏egg production (line 535-536, [200]).

We believe that an expansion of the text with regards to egg production in EU, at this stage of the MS is less feasable, and especially since after a very comprehensive literature review we found only two relevant projects. Therefore, to ensure that the reader understands that we are not dealing with e.g. turkeys, quails etc but broilers, we have implemented several measures:

  1. We solely useת everywhere in the text where it is relevantת the concept of "broiler chickens" and "broiler production" and not use "poultry production". (Lines 42, 400, 555).
  2. We add in Conclusion: "We recommend the development of enhanced research infrastructures in the sectors of laying hens, turkeys and quails etc., on the basis of their being underrepresented in the plethora of active research projects in Europe". (Lines 557-560).

We hope this satisfies and clarifies this important insight.